

**COMPUTO**

**ISSN 2824-7795**

# `regMMD`: an `R` package for parametric estimation and regression with maximum mean discrepancy

**The `R` package `regMMD`**

Pierre Alquier [1]  Department of Information Systems, Data Analytics and Operations, ESSEC Business School

Mathieu Gerber   School of Mathematics, University of Bristol

Date published: 2025-05-07    Last modified: 2025-05-07

**Abstract**

The Maximum Mean Discrepancy (MMD) is a kernel-based metric widely used for non-parametric tests and estimation. Recently, it has also been studied as an objective function for parametric estimation, as it has been shown to yield robust estimators. We have implemented MMD minimization for parameter inference in a wide range of statistical models, including various regression models, within an R package called `regMMD`. This paper provides an introduction to the `regMMD` package. We describe the available kernels and optimization procedures, as well as the default settings. Detailed applications to simulated and real data are provided.

*Keywords:* parameter estimation, regression, robust statistics, minimum distance estimation, kernel methods, maximum mean discrepancy

# Contents

[1]Corresponding author: pierre.alquier@essec.edu

# 1 Introduction

In some models, popular estimators such as the maximum likelihood estimator (MLE) can become very unstable in the presence of outliers. This has motivated research into robust estimation procedures that would not suffer from this issue. Various notions of robustness have been proposed, depending on the type of contamination in the data. Notably, the Huber contamination model (Huber 1992) considers random outliers while, more recently, stricter notions have been proposed to ensure robustness against adversarial contamination of the data.

The maximum mean discrepancy (MMD) is a kernel-based metric that has received considerable attention in the past 15 years. It allows for the development of tools for nonparametric tests and estimation (Chwialkowski, Strathmann, and Gretton 2016; Gretton et al. 2007). We refer the reader to (Muandet et al. 2017) for a comprehensive introduction to MMD and its applications. A recent series of papers has suggested that minimum distance estimators (MDEs) based on the MMD are robust to both Huber and adversarial contamination. These estimators were initially proposed for training generative AI (Dziugaite, Roy, and Ghahramani 2015; Sutherland et al. 2016; Li et al. 2017), and the study of their statistical properties for parametric estimation, including robustness, was initiated by (Briol et al. 2019; Chérief-Abdellatif and Alquier 2022; Alquier and Gerber 2024). We also point out that the MMD has been successfully used to define robust estimators that are not MDEs, such as bootstrap methods based on MMD (Dellaporta et al. 2022) or Approximate Bayesian Computation (Legramanti, Durante, and Alquier 2025).

Unfortunately, we are not aware of any software that allows for the computation of MMD-based MDEs. To make these tools available to the statistical community, we developed the R package called regMMD. This package allows for the minimization of the MMD distance between the empirical distribution and a statistical model. Various parametric models can be fitted, including continuous distributions such as Gaussian and gamma, and discrete distributions such as Poisson and binomial. Many regression models are also available, including linear, logistic, and gamma regression. The regMMD package is available on the CRAN website (R Core Team 2020): [regMMD page](regMMD page)

The optimization is based on the strategies proposed by (Briol et al. 2019; Chérief-Abdellatif and Alquier 2022; Alquier and Gerber 2024). For some models we have an explicit formula for the gradient of the MMD, in which case we use gradient descent, see e.g. (Boyd and Vandenberghe 2004; Nesterov 2018) to perform the optimization. For most models such a formula does not exist, but we can however approximate the gradient without bias by Monte Carlo sampling. This allows to use the

stochastic gradient algorithm of (Robbins and Monro 1951), that is one of the most popular estimation methods in machine learning (Bottou 2004). We refer to the reader to (Wright 2018; J. C. Duchi 2018) and Chapter 5 in (Bach 2024) for comprehensive introductions to otpimization for statistics and machine learning, including stochastic optimization methods.

The paper is organized as follows. In Section 2 we briefly recall the construction of the MMD metric and the MDE estimators based on the MMD. In Section 3 we detail the content of the package regMMD: the available models and kernels, and the optimization procedures used in each case. Finally, in Section 4 we provide examples of applications of regMMD. Note that these experiments are not meant to be a comprehensive comparison of MMD to other robust estimation procedures. Exhaustive comparisons can be found in (Briol et al. 2019; Chérief-Abdellatif and Alquier 2022; Alquier and Gerber 2024). The objective is simply to illustrate the use of regMMD through pedagogical examples.

## 2 Statistical background

### 2.1 Parametric estimation

Let $X_1, \ldots, X_n$ be $\mathcal{X}$-valued random variables identically distributed according to some probability distribution $P^0$, and let $(P_\theta, \theta \in \Theta)$ be a statistical model. Given a metric $d$ on probability distributions, we are looking for an estimator of $\theta_0 \in \arg\min_{\theta \in \Theta} d(P_\theta, P^0)$ when such a minimum exists. Letting $\hat{P}_n = \frac{1}{n} \sum_{i=1}^{n} \delta_{X_i}$ denote the empirical probability distribution, the minimum distance estimator (MDE) $\hat{\theta}$ is defined as follows:

$$\hat{\theta} \in \arg\min_{\theta \in \Theta} d(P_\theta, \hat{P}_n).$$

The robustness properties of MDEs for well chosen distances was studied as early as in (Wolfowitz 1957; Parr and Schucany 1980; Yatracos 2022). When $d(P_\theta, \hat{P}_n)$ has no minimum, the definition can be replaced by an $\varepsilon$-approximate minimizer, without consequences on the the properties of the estimator, as shown in to (Briol et al. 2019; Chérief-Abdellatif and Alquier 2022).

Let $\mathcal{H}$ be a Hilbert space, let $\| \cdot \|_{\mathcal{H}}$ and $\langle \cdot, \cdot \rangle_{\mathcal{H}}$ denote the associated norms and scalar products, respectively, and let $\varphi : \mathcal{X} \to \mathcal{H}$. Then, for any probability distribution $P$ on $\mathcal{X}$ such that $\mathbb{E}_{X \sim P}[\|\varphi(X)\|_{\mathcal{H}}] < +\infty$ we can define the mean embedding $\mu(P) = \mathbb{E}_{X \sim P}[\varphi(X)]$. When the mean embedding $\mu(P)$ is defined for any probability distribution $P$ (e.g. because the map $\varphi$ is bounded in $\mathcal{H}$), for any probability distributions $P$ and $Q$ we put

$$\mathbb{D}(P, Q) := \|\mu(P) - \mu(Q)\|_{\mathcal{H}}.$$

Letting $k(x, y) = \langle \varphi(x), \varphi(y) \rangle_{\mathcal{H}}$, it appears that $\mathbb{D}(P, Q)$ depends on $\varphi$ only through $k$, as we can rewrite:

$$\mathbb{D}^2(P, Q) = \mathbb{E}_{X, X' \sim P}[k(X, X')] - 2\mathbb{E}_{X \sim P, X' \sim Q}[k(X, X')] + \mathbb{E}_{X, X' \sim Q}[k(X, X')]. \tag{1}$$

When $\mathcal{H}$ is actually a RKHS for the kernel $k$ (see (Muandet et al. 2017) for a definition), $\mathbb{D}(P, Q)$ is called the maximum mean discrepancy (MMD) between $P$ and $Q$. A condition on $k$ (universal kernel) ensures that the map $\mu$ is injective, and thus that $\mathbb{D}$ satisfies the axioms of a metric. Examples of universal kernels are known, such as the Gaussian kernel $k(x, y) = \exp(-\|x - y\|^2/\gamma^2)$ or the Laplace kernel $k(x, y) = \exp(-\|x - y\|/\gamma)$, see (Muandet et al. 2017) for more examples and references to the proofs.

The properties of MDEs based on MMD were studied in (Briol et al. 2019; Chérief-Abdellatif and Alquier 2022). In particular, when the kernel $k$ is bounded, this estimator enjoys very strong robustness properties. We cite the following very simple result.

**Theorem 2.1** (special case of Theorem 3.1 in (Chérief-Abdellatif and Alquier 2022)). *Assume $X_1, \dots, X_n$ are i.i.d. from $P^0$. Assume the kernel $k$ is bounded by 1. Then*

$$\mathbb{E}\left[\mathbb{D}\left(P_{\hat{\theta}}, P^0\right)\right] \leq \inf_{\theta \in \Theta} \mathbb{D}\left(P_\theta, P^0\right) + \frac{2}{\sqrt{n}}.$$

Additional non-asymptotic results can be found in (Briol et al. 2019; Chérief-Abdellatif and Alquier 2022). In particular, Theorem 3.1 of (Chérief-Abdellatif and Alquier 2022) also covers non independent observations (time series). An asymptotic study of $\hat{\theta}$, including conditions for asymptotic normality, can be found in (Briol et al. 2019). All these works provide strong theoretical evidence that $\hat{\theta}$ is very robust to random and adversarial contamination of the data, and this is supported by empirical evidence.

## 2.2 Regression

Let us now consider a regression setting: we observe $(X_1, Y_1), \dots, (X_n, Y_n)$ in $\mathcal{X} \times \mathcal{Y}$ and we want to estimate the conditional distribution $P^0_{Y|X=x}$ of $Y$ given $X = x$ for any $x$. %A direct application of the method in the previous section to the random variables $(X_1, Y_1), \dots, (X_n, Y_n)$ would lead to the estimation of the joint distribution of the pair $(X, Y)$, which is not the objective of regression models. To this end we consider a statistical model $(P_\beta, \beta \in \mathcal{B})$ and model the conditional distribution of $Y$ given $X = x$ by $(P_{\beta(x,\theta)}, \theta \in \Theta)$ where $\beta(\cdot, \cdot)$ is a specified function $\mathcal{X} \times \Theta \to \mathcal{B}$. The first estimator proposed by (Alquier and Gerber 2024) is:

$$\hat{\theta}_{\text{reg}} \in \arg\min_{\theta \in \Theta} \mathbb{D}\left(\frac{1}{n}\sum_{i=1}^n \delta_{X_i} \otimes P_{\beta(X_i,\theta)}, \frac{1}{n}\sum_{i=1}^n \delta_{X_i} \otimes \delta_{Y_i}\right)$$

where $\mathbb{D}$ is the MMD defined by a product kernel, that is a kernel of the form $k((x,y),(x',y')) = k_X(x,x')k_Y(y,y')$ (non-product kernels are theoretically possible, but not implemented in the package). Asymptotic and non-asymptotic properties of $\hat{\theta}$ are studied in (Alquier and Gerber 2024). The computation of $\hat{\theta}$ is however slow when the sample size $n$ is large, as it can be shown that the criterion defining this estimator is the sum of $n^2$ terms.

By contrast, the following alternative estimator

$$\tilde{\theta}_{\text{reg}} \in \arg\min_{\theta \in \Theta} \frac{1}{n}\sum_{i=1}^n \mathbb{D}\left(\delta_{X_i} \otimes P_{\beta(X_i,\theta)}, \delta_{X_i} \otimes \delta_{Y_i}\right),$$

has the advantage to be defined through a criterion which is a sum of only $n$ terms. Intuitively, this estimator can be interpreted as a special case of $\hat{\theta}_{\text{reg}}$ where $k_X(x,x') = \mathbf{1}_{\{x=x'\}}$. An asymptotic study of $\tilde{\theta}_{\text{reg}}$ is provided by (Alquier and Gerber 2024). The theory and the experiments suggest that both estimators are robust, but that $\hat{\theta}_{\text{reg}}$ is more robust than $\tilde{\theta}_{\text{reg}}$. However, for computations reasons, for large sample sizes $n$ ($n > 5\,000$, say) only the latter estimator can be computed in a reasonable amount of time.

## 3 Package content and implementation

The package regMMD allows to compute the above estimators in a large number of classical models. We first provide an overview of the two main functions with their default settings: mmd_reg for regression, and mmd_est for parameter estimation. We then give some details on their implementations and options. These functions have many options related to the choice of kernels, the choice of the bandwidth parameters and of the parameters of the optimization algorithms used to compute the estimators. To save space, in this section we only discuss the options that are fundamental from a

130 statistical perspective. We refer the reader to the package documentation for a full description of the
131 available options.

132 We start by loading the package and fixing the seed to ensure reproducibility.

```
require("regMMD")
```

133 Loading required package: regMMD

```
set.seed(0)
```

## 3.1 Overview of the function `mmd_est`

135 The function `mmd_est` performs parametric estimation as described in Section 2.1. Its required
136 arguments are the data x and the type of model `model` (see Section 3.5 for the list of available models).
137 Each model implemented in the package has one or two parameters, namely `par1` and `par2`. If the
138 model contains a parameter that is fixed (i.e. not estimated from the data) then its value must be
139 specified by the user. On the other hand, a value for a parameter that we want to estimate from
140 the data does not have to be given as an input. If, however, a value is provided then it is used to
141 initialize the optimization algorithm that serves at computing the estimator (see below). Otherwise
142 an initialization by default is used.

143 For example, there are three Gaussian univariate models: `Gaussian.loc`, `Gaussian.scale` and
144 `Gaussian`. In each model `par1` is the mean and `par2` is the standard deviation. We will use the
145 following data:

```
x = rnorm(100,1.9,1)
hist(x)
```

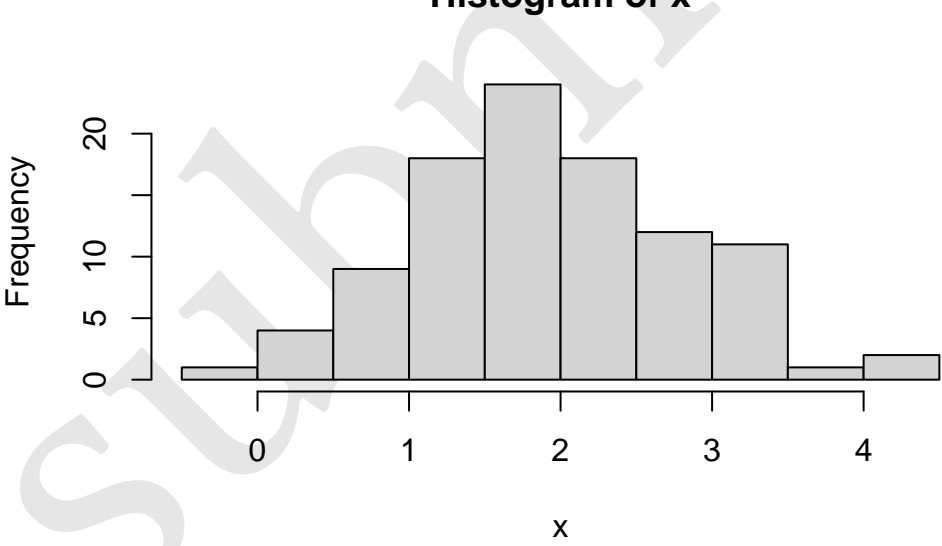

**Histogram of x**

146

147 In the `Gaussian` model the two parameters `par1` and `par2` are estimated from the data and the MMD
148 estimator of $\theta$ =(par1,par1) can be computed as follows:

```
estim = mmd_est(x,model="Gaussian")
```

149 Here, we mention that the output of both `mmd_est` and `mmd_reg` is a list that contains error messages
150 (if any), the estimated parameters (if no error prevented their computations) and various information

on the model, the estimators and the algorithms. The major information can be retrieved through a summary function:

```
summary(estim)
```

```
==================== Summary ======================
Model:                Gaussian
-----------------------------------------------------------
Algorithm:            SGD
Kernel:               Gaussian
Bandwidth:            0.856108154564806
-----------------------------------------------------------
Parameters:

par1: mean -- initialized at 1.867
      estimated value: 1.8759

par2: standard deviation -- initialized at 0.7786
      estimated value: 0.8889
===========================================================
```

Note that some of the information provided here (like the algorithm used) will be detailed below.

Still in the Gaussian model, if we enter

```
estim = mmd_est(x,model="Gaussian",par1=0,par2=1)
summary(estim)
```

```
==================== Summary ======================
Model:                Gaussian
-----------------------------------------------------------
Algorithm:            SGD
Kernel:               Gaussian
Bandwidth:            0.856108154564806
-----------------------------------------------------------
Parameters:

par1: mean -- initialized at 0
      estimated value: 1.8758

par2: standard deviation -- initialized at 1
      estimated value: 0.8862
===========================================================
```

we simply enforce the optimization algorithm that serves at computing the estimator to use $\theta_0 = (0, 1)$ as starting value. In the Gaussian.loc model only the location parameter (the mean) is estimated. Thus, to compute $\theta = $ par1 it is necessary to specify the standard deviation par2. For instance,

```
estim = mmd_est(x,model="Gaussian.loc",par2=1)
summary(estim)
```

```
==================== Summary ======================
Model:                Gaussian.loc
-----------------------------------------------------------
```

```
191  Algorithm:            GD
192  Kernel:               Gaussian
193  Bandwidth:            0.856108154564806
194  ------------------------------------------------------------
195  Parameters:
196
197  par1: mean -- initialized at 1.867
198        estimated value: 1.8817
199
200  par2: standard deviation -- fixed by user: 1
201  ============================================================
```

will estimate $\theta$ in the model $\mathcal{N}(\theta, 1)$. If we provide a value for par1 then the optimization algorithm used to compute $\hat{\theta}$ will use $\theta_0 =$ par1 as starting value. Finally, In the Gaussian.scale model only the scale parameter (standard deviation) is estimated. That is, to estimate $\theta =$ par2 in e.g. the $\mathcal{N}(2, \theta^2)$ distribution we can use

```
estim = mmd_est(x,model="Gaussian.scale",par1=2)
summary(estim)
```

```
206  ====================== Summary ========================
207  Model:                Gaussian.scale
208  ------------------------------------------------------------
209  Algorithm:            SGD
210  Kernel:               Gaussian
211  Bandwidth:            0.856108154564806
212  ------------------------------------------------------------
213  Parameters:
214
215  par1: mean -- fixed by user: 2
216
217  par2: standard deviation -- initialized at 0.6981
218        estimated value: 0.9005
219  ============================================================
```

## 3.2   Overview of the function mmd_reg

The function mmd_reg is used for regression models (Section 2.2) and requires to specify two arguments, namely the output vector y of size $n$ and the $n \times q$ input matrix X. By default, the functions performs linear regression with Gaussian error noise (with unknown variance) and the regression model to be used can be changed through the option model of mmd_reg (see Section 3.5 for the list of available models). In addition, by default, if the input matrix X contains no column whose entries are all equal then an intercept is added to the model, that is, a column of 1's is added to X. This default setting can be disabled by setting the option intercept of mmd_reg to FALSE.

All regression models implemented in the package have a parameter par1, and some of them have an additional scalar parameter par2. If a model has par1 as unique parameter then the parameter to be estimated is $\theta =$ par1 and the conditional distribution of $Y$ given $X = x$ is modelled using a model of the form $(P_{\beta(x^\top \theta)}, \theta \in \mathbb{R}^k)$, with $k$ the size of $x$. For instance, for Poisson regression the distribution $P_{\beta(x^\top \theta)}$ is the Poisson distribution with mean $\exp(x^\top \theta)$. By contrast, some models have an additional parameter par2 that also needs to be estimated from the data, so that $\theta =$(par1,par2). For these models the conditional distribution of $Y$ given $X = x$ is modeled using a model of the form

$(P_{\beta(x^\top \gamma, \psi)}, \theta = (\gamma, \psi) \in \mathbb{R}^{k+1})$. For instance, for the Gaussian linear regression model with unknown variance $P_{\beta(x^\top \gamma, \psi)} = \mathcal{N}(x^\top \gamma, \psi^2)$. Finally, some models have an additional parameter par2 whose value needs to be specified by the user. For these models $\theta$ =par1, par2= $\psi$ for some $\psi$ fixed by the user, and the conditional distribution of $Y$ given $X = x$ is modeled using a model of the form $(P_{\beta(x^\top \theta, \psi)}, \theta \in \mathbb{R}^k)$. For instance, for the Gaussian linear regression model with known variance, $P_{\beta(x^\top \theta, \psi)} = \mathcal{N}(x^\top \theta, \psi^2)$. Remark that for all models implemented in the package par1 is therefore the vector of regression coefficients. As with the function mmd_est, if a value for a parameter that needs to be estimated from the data is provided then it is used to initialize the optimization algorithm that serves at computing the estimator, otherwise an initialization by default is used. It is important to note that the number $k$ of regression coefficients of a model is either $q$ (the number of columns of the input matrix X) or $q + 1$ if an intercept has been added by mmd_reg. In the latter case, if we want to provide a value for par1 then it must be vector of length $q$.

For example, there are two linear regression model with Gaussian noise: linearGaussian which assumes that noise variance is unknown and linearGaussian.loc which assumes that noise variance is unknown. By default, the former model is used by mmd_reg, and thus linear regression can be simply performed using estim = mmd_reg(y,X). By default mmd_reg uses the MMD estimator $\tilde{\theta}_{\text{reg}}$, which we recall is cheaper to compute that the alternative MMD estimator $\hat{\theta}_{\text{reg}}$.

## 3.3 Kernels and bandwidth parameters

For parametric models (Section 2.1) the MMD estimator of $\theta$ is computed with a kernel $k(x, x')$ of the form $k(x, x') = K(\|x - x'\|/\gamma)$ for some bandwidth parameter $\gamma > 0$ and some function $K : [0, \infty) \to [0, \infty)$. The choice of $K$ and $\gamma$ can be specified through the option kernel and bdwth of the function mmd_est, respectively. By default, the median heuristic is used to choose $\gamma$ (the median heuristic was used successfully in many applications of MMD, for example (Gretton et al. 2012), see also (Garreau, Jitkrittum, and Kanagawa 2017) for a theoretical analysis). The following three options are available for the function $K$:

- Gaussian: $K(u) = \exp(-u^2)$,
- Laplace: $K(u) = \exp(-u)$,
- Cauchy: $K(u) = 1/(2 + u^2)$.

Similarly, for regression models (Section 2.2), the MMD estimators are computed with $k_X(x, x') = K_X(\|x - x'\|/\gamma_X)$ and $k_Y(y, y') = K_Y(\|y - y'\|/\gamma_Y)$ for some bandwidth parameters $\gamma_X \geq 0$ and $\gamma_Y > 0$ and functions $K_X, K_Y : [0, \infty) \to [0, \infty)$. The choice of $K_X$, $K_Y$, and $\gamma_X$ and $\gamma_Y$ can be specified through the option kernel.x, kernel.y, bdwth.x and bdwth.y of the function mmd_reg, respectively. The available choices for $K_X$ and $K_Y$ are the same as for $K$, and by default the median heuristic is used to select $\gamma_Y$. By default, bdwth.x=0 and thus it is the estimator $\tilde{\theta}_{\text{reg}}$ that is used by mmd_reg. The alternative estimator $\hat{\theta}_{\text{reg}}$ can be computed either by providing a positive value for bdwth.x or by setting bdwth.x="auto", in which case a rescaled version of the median heuristic is used to choose a positive value for $\gamma_X$.

## 3.4 Optimization methods

Depending on the model, the package regMMD uses either gradient descent (GD) or stochastic gradient descent (SGD) to compute the estimators.

More precisely, for parametric estimation it is proven in Section 5 of (Chérief-Abdellatif and Alquier 2022) that the gradient of $\mathbb{D}^2(P_\theta, \hat{P}_n)$ with respect to $\theta$ is given by

$$\nabla_\theta \mathbb{D}^2(P_\theta, \hat{P}_n) = 2\mathbb{E}_{X,X' \sim P_\theta}\left[\left(k(X, X') - \frac{1}{n}\sum_{i=1}^n k(X_i, X)\right)\nabla_\theta[\log p_\theta(X)]\right] \tag{2}$$

under suitable assumptions on $P_\theta$, including the existence of a density $p_\theta(X)$ and its differentiability. In some models there is an explicit formula for the expectation in Equation 2. This is for instance the case for the Gaussian mean model, and for such models a gradient descent algorithm is used to compute the MMD estimator. For models where we cannot compute explicitly the expectation in Equation 2 it is possible to compute an unbiased estimate of the gradient by sampling from $P_\theta$. In this scenario the MMD estimator is computed using AdaGrad (J. Duchi, Hazan, and Singer 2011), and adaptive step-size SGD algorithm. Finally, in very specific models, $\mathbb{D}(P_\theta, \hat{P}_n)$ can be evaluated explicitly in which case we can perform exact optimization `exact`. This is for example the case when $P_\theta$ is the (discrete) uniform distribution on $\{1, 2, \ldots, \theta\}$.

In `mmd_est`, for each model all the available methods for computing the estimator are implemented. The method used by default is chosen according to the ranking: `exact>GD>SGD`. We can enforce another method with the `method` option. For instance, `estim <- mmd_est(x,model="Gaussian")` is equivalent to `estim <- mmd_est(x,model="Gaussian",method="GD")` and we can enforce the use of SGD with `estim <- mmd_est(x,model="Gaussian",method="SGD")`.

For regression models, formulas similar to the one given in Equation 2 for the gradient of the criteria defining the estimators $\hat{\theta}_{\mathrm{reg}}$ and $\tilde{\theta}_{\mathrm{reg}}$ are provided in Section S1 of the supplement of (Alquier and Gerber 2024). For the two estimators, this gradient can be computed explicitly for all linear regression models when `kernel.y="Gaussian"` and for the logistic regression model.

For the estimator $\tilde{\theta}_{\mathrm{reg}}$, and as for parametric estimation, gradient descent is used when the gradient of the objective function can be computed explicitly, and otherwise the optimization is performed using Adagrad. In `mmd_reg` gradient descent is implemented using backtracking line search to select the step-size to use at each iteration, and a stopping rule is implemented to stop the optimization earlier when possible.

The computation of $\hat{\theta}_{\mathrm{reg}}$ is more delicate. Indeed, the objective function defining this estimator is the sum of $n^2$ terms (see Section 2.2), implying that minimizing this function using gradient descent or SGD leads to algorithms for computing the estimator that require $\mathcal{O}(n^2)$ operations per iteration. In the package, to reduce the cost per iteration we implement the strategy proposed in Section S1 of the supplement of (Alquier and Gerber 2024). Importantly, with this strategy the optimization is performed using an unbiased estimate of the gradient of the objective function, even when the gradient of the $n^2$ terms of objective function can be computed explicitly. It is however possible to use the explicit formula for these gradients to reduce the variance of the noisy gradient, which we do when possible. As for the computation of $\tilde{\theta}_{\mathrm{reg}}$ a stopping rule is implemented to stop the optimization earlier when possible. With this package it is feasible to compute $\hat{\theta}_{\mathrm{reg}}$ in a reasonable amount of time for dataset containing up to a few thousands data points (up-to $n \approx 5\,000$ observations, say).

Finally, we stress that, from a computational point of view, MMD estimation in regression models is a much more challenging task than in parametric models for at least two reasons. Firstly, while in the latter task the dimension of the parameter of interest $\theta$ is at most two for the models implemented in this package (see below), in regression the dimension of $\theta$ can be much larger, depending on the number of explanatory variables. Secondly, the objective functions to minimize for regression models are ''more non-linear''. When estimating a regression model it is therefore a good practice to verify that the optimization of the objective function has converged. This can be done by inspecting the sequence of $\theta$ values computed by the optimization algorithm, accessible from the object `trace` of `mmd_reg`.

## 3.5 Available models

List of univariate models in `mmd_est`:

- Gaussian $\mathcal{N}(m, \sigma^2)$: `Gaussian`(estimation of $m$ and $\sigma$), `Gaussian.loc`(estimation of $m$) and `Gaussian.scale`(estimation of $\sigma$),
- Cauchy: `Cauchy`(estimation of the location parameter),
- Pareto: `Pareto`(estimation of the exponent),
- exponential $\mathscr{E}(\lambda)$: 'exponential},
- gamma $Gamma(a, b)$: `gamma`(estimation of $a$ and $b$), `gamma.shape`(estimation of $a$) and `gamma.rate`(estimation of $b$),
- continuous uniform: `continuous.uniform.loc`(estimation of $m$ in $\mathscr{U}[m - \frac{L}{2}, m + \frac{L}{2}]$, where $L$ is fixed by the user), `continuous.uniform.upper`(estimation of $b$ in $\mathscr{U}[a, b]$) and `continuous.uniform.lower.upper`(estimation of both $a$ and $b$ in $\mathscr{U}[a, b]$),
- Dirac $\delta_a$: `Dirac`(estimation of $a$; while this might sound uninteresting at first, this can be used to define a ''model-free'' robust location parameter),
- discrete uniform $\mathscr{U}(\{1, 2, \dots, N\})$: `discrete.uniform`(estimation of $N$),
- binomial $Bin(N, p)$: `binomial`(estimation of $N$ and $p$), `binomial.size`(estimation of $N$) and `binomial.prob`(estimation of $p$),
- geometric $\mathscr{G}(p)$: `geometric`,
- Poisson $\mathscr{P}(\lambda)$: `Poisson`.

List of multivariate models in `mmd_est`:

- multivariate Gaussian $\mathcal{N}(\mu, UU^T)$: `multidim.Gaussian` (estimation of $\mu$ and $U$), `multidim.Gaussian.loc` (estimation of $\mu$ while $U = \sigma I$ for a fixed $\sigma > 0$) and `multidim.Gaussian.scale` (estimation of $U$ while $\mu$ is fixed),
- Dirac mass $\delta_a$: `multidim.Dirac`.

List of regression models in `mmd_reg`:

- linear regression models with Gaussian noise: `linearGaussian` (unknown noise variance) and `linearGaussian.loc` (known noise variance),
- exponential regression: `exponential`,
- gamma regression: `gamma`, or `gamma.loc` when the precision parameter is known,
- beta regression: `beta`, or `beta.loc` when the precision parameter is known,
- logistic regression: `logistic`,
- Poisson regression: `poisson`.

# 4 Detailed examples

## 4.1 Toy example: robust estimation in the univariate Gaussian model

We start with a very simple illustration on synthetic data. We choose one of the simplest model, namely, estimation of the mean of a univariate Gaussian random variable. The statistical model is $\mathcal{N}(\theta, 1)$, which is the `Gaussian.loc` model in the package. We remind the above, using the default settings:

```
estim = mmd_est(x,model="Gaussian.loc",par2=1)
summary(estim)
```

```
====================== Summary ======================
Model:              Gaussian.loc
-----------------------------------------------------
Algorithm:          GD
Kernel:             Gaussian
Bandwidth:          0.856108154564806
```

```
364   ----------------------------------------------------------
365   Parameters:
366
367   par1: mean -- initialized at 1.867
368         estimated value: 1.8817
369
370   par2: standard deviation -- fixed by user: 1
371   ==========================================================
```

The user can also impose a different bandwidth and kernel, which will result in a different estimator:

```
estim =  mmd_est(x,model="Gaussian.loc",par2=1,bdwth=0.6,kernel="Laplace")
summary(estim)
```

```
373   ===================== Summary =====================
374   Model:               Gaussian.loc
375   ----------------------------------------------------------
376   Algorithm:           GD
377   Kernel:              Laplace
378   Bandwidth:           0.6
379   ----------------------------------------------------------
380   Parameters:
381
382   par1: mean -- initialized at 1.867
383         estimated value: 1.8795
384
385   par2: standard deviation -- fixed by user: 1
386   ==========================================================
```

We end up the discussion on the Gaussian mean example by a toy experiment: we replicate $N = 200$ times the simulation of $n = 100$ i.i.d. random variables $\mathcal{N}(-2, 1)$ and compare the maximum likelihood estimator (MLE), equal to the empirical mean, the median, and the MMD estimator with a Gaussian and with a Laplace kernel, using in both cases the median heuristic for the bandwidth parameter. We report the Mean Absolute Error over all the simulations (MAE) as well as the standard deviation of the Absolute Error (sdAE).

```
N = 200
n = 100
theta = -2
err = matrix(data=0,nr=N,nc=4)
for (i in 1:N)
{
    X = rnorm(n,theta,1)
    thetamle = mean(X)
    thetamed = median(X)
    estim = mmd_est(X,model="Gaussian.loc",par2=1)
    thetanew1 = estim$estimator
    estim = mmd_est(X,model="Gaussian.loc",par2=1,kernel="Laplace")
    thetanew2 = estim$estimator
    err[i,] = abs(theta-c(thetamle,thetanew1,thetanew2,thetamed))
}
results = matrix(0,nr=2,nc=4)
```

```r
results[1,] = colSums(err)/N
for (i in 1:4) results[2,i] = sd(err[,i])
colnames(results) = c('MLE','MMD (Gaussian kernel)','MMD (Laplace kernel)','Median')
rownames(results) = c('MAE','sdAE')
final = as.table(results)
knitr::kable(final,caption="Estimation of the mean in the uncontaminated case")
```

Table 1: Estimation of the mean in the uncontaminated case

|      | MLE       | MMD (Gaussian kernel) | MMD (Laplace kernel) | Median    |
|------|-----------|-----------------------|----------------------|-----------|
| MAE  | 0.0780971 | 0.0905728             | 0.0878320            | 0.1038774 |
| sdAE | 0.0578317 | 0.0658958             | 0.0640874            | 0.0761439 |

In a second time, we repeat the same experiment with contamination: two of the $X_i$'s are sampled from a standard Cauchy distribution instead.

```r
N = 200
n = 100
theta = -2
err = matrix(data=0,nr=N,nc=4)
for (i in 1:N)
{
    cont = rcauchy(2)
    X = c(rnorm(n-2,theta,1),cont)
    thetamle = mean(X)
    thetamed = median(X)
    estim = mmd_est(X,model="Gaussian.loc",par2=1)
    thetanew1 = estim$estimator
    estim = mmd_est(X,model="Gaussian.loc",par2=1,kernel="Laplace")
    thetanew2 = estim$estimator
    err[i,] = abs(theta-c(thetamle,thetanew1,thetanew2,thetamed))
}
results = matrix(0,nr=2,nc=4)
results[1,] = colSums(err)/N
for (i in 1:4) results[2,i] = sd(err[,i])
colnames(results) = c('MLE','MMD (Gaussian kernel)','MMD (Laplace kernel)','Median')
rownames(results) = c('MAE','sdAE')
final = as.table(results)
knitr::kable(final,caption="Estimation of the mean in the contaminated case")
```

Table 2: Estimation of the mean in the contaminated case

|      | MLE       | MMD (Gaussian kernel) | MMD (Laplace kernel) | Median    |
|------|-----------|-----------------------|----------------------|-----------|
| MAE  | 0.1411296 | 0.0964510             | 0.0941981            | 0.1061498 |
| sdAE | 0.2853344 | 0.0734671             | 0.0708889            | 0.0772327 |

The results are as expected: under no contamination, the MLE is known to be efficient and is therefore the best estimator. On the other hand, the MLE (i.e. the empirical mean) is known to be very sensitive to outliers. In contrast, the MMD estimators and the median are robust estimators of $\theta$. We refer the

reader to (Briol et al. 2019; Chérief-Abdellatif and Alquier 2022) for more discussions and experiments on the robustness MMD estimators.

Note that while the median is a natural robust alternative to the MLE in the Gaussian mean model, such an alternative is not always available. Consider for example the estimation of the standard deviation of a Gaussian, with known zero mean: $\mathcal{N}(0,\theta^2)$. We cannot use (directly) a median in this case, while the MMD estimators are available. We repeat similar experiments as above in this model. We let $N$ and $n$ be as above and sample the uncontaminated observations from the $\mathcal{N}(0,1)$ distributions. We them compare the MLE to the MMD with Gaussian and Laplace kernel both in the uncontaminated case and under Cauchy contamination for two observations. The conclusions are completely similar to the ones obtained in the Gaussian location experiments.

```
N = 200
n = 100
theta=1
err = matrix(data=0,nr=N,nc=3)
for (i in 1:N)
{
    X = rnorm(n,0,theta)
    thetamle = sqrt(mean(X^2))
    estim = mmd_est(X,model="Gaussian.scale",par1=0)
    thetanew1 = estim$estimator
    estim = mmd_est(X,model="Gaussian.scale",par1=0,kernel="Laplace")
    thetanew2 = estim$estimator
    err[i,] = abs(theta-c(thetamle,thetanew1,thetanew2))
}
results = matrix(0,nr=2,nc=3)
results[1,] = colSums(err)/N
for (i in 1:3) results[2,i] = sd(err[,i])
colnames(results) = c('MLE','MMD (Gaussian kernel)','MMD (Laplace kernel)')
rownames(results) = c('MAE','sdAE')
final = as.table(results)
knitr::kable(final,caption="Estimation of the standard-deviation in the uncontaminated case")
```

Table 3: Estimation of the standard-deviation in the uncontaminated case

|      | MLE       | MMD (Gaussian kernel) | MMD (Laplace kernel) |
|------|-----------|-----------------------|----------------------|
| MAE  | 0.0530777 | 0.0649238             | 0.0636323            |
| sdAE | 0.0390381 | 0.0507710             | 0.0497721            |

```
N = 200
n = 100
theta=1
err = matrix(data=0,nr=N,nc=3)
for (i in 1:N)
{
    cont = rcauchy(2)
    X = c(rnorm(n-2,0,theta),cont)
    thetamle = sqrt(mean(X^2))
    estim = mmd_est(X,model="Gaussian.scale",par1=0)
    thetanew1 = estim$estimator
```

```
    estim = mmd_est(X,model="Gaussian.scale",par1=0,kernel="Laplace")
    thetanew2 = estim$estimator
    err[i,] = abs(theta-c(thetamle,thetanew1,thetanew2))
}
results = matrix(0,nr=2,nc=3)
results[1,] = colSums(err)/N
for (i in 1:3) results[2,i] = sd(err[,i])
colnames(results) = c('MLE','MMD (Gaussian kernel)','MMD (Laplace kernel)')
rownames(results) = c('MAE','sdAE')
final = as.table(results)
knitr::kable(final,caption="Estimation of the standard-deviation in the contaminated case")
```

Table 4: Estimation of the standard-deviation in the contaminated case

|      | MLE       | MMD (Gaussian kernel) | MMD (Laplace kernel) |
|------|-----------|-----------------------|----------------------|
| MAE  | 0.4608255 | 0.0678643             | 0.0668497            |
| sdAE | 1.7910074 | 0.0503154             | 0.0495938            |

## 4.2  Robust linear regression

### 4.2.1  Dataset and model

To illustrate the use of the regMMD package to perform robust linear regression we use the R built-in dataset airquality. The dataset contains daily measurements of four variables related to air quality in New York, namely the ozone concentration (variable Ozone), the temperature (variable Temp), the wind speed (variable Wind) and the solar radiation (variable Solar.R). Observations are reported for the period ranging from the 1st of May 1973 to the 30th of September 1973, resulting in a total of 153 observations. The dataset contains 42 observations with missing values, which we remove from the sample.

```
air.data = na.omit(airquality)
hist(air.data$Ozone,breaks=20)
```

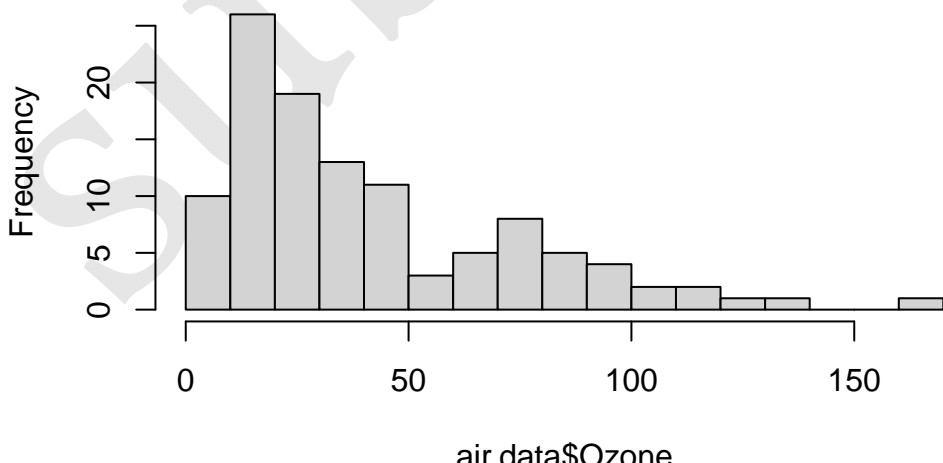

418 We report the histogram of the log of the variable `Ozone`:

```
hist(log(air.data$Ozone),breaks=20)
```

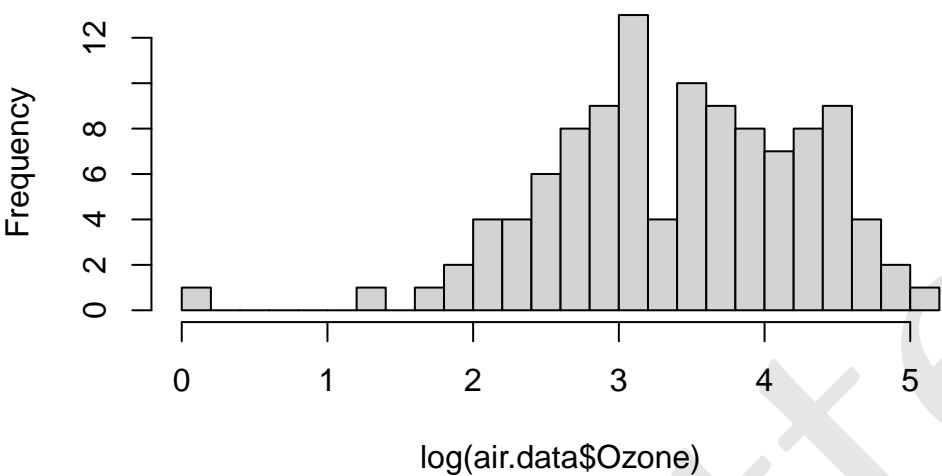

**Histogram of log(air.data$Ozone)**

419

420 From this plot we observe that there is one isolated observation (i.e. outlier), for which the log of the
421 observed value of `Ozone` is 0.

422 Our aim is to study the link between the level of ozone concentration and the other three variables
423 present in the dataset using the following linear regression model:

$$\log(\texttt{Ozone}) = \alpha + \beta_1 \texttt{Temp} + \beta_2 (\texttt{Temp})^2 + \beta_3 \texttt{Wind} + \beta_4 (\texttt{Wind})^2$$
$$+ \beta_5 \texttt{Solar.R} + \beta_6 (\texttt{Solar.R})^2 + \epsilon$$

424 where $\epsilon \sim \mathcal{N}(0, \sigma^2)$. The noise variance $\sigma^2$ is assumed to be unknown so that the model contains 8
425 parameters that need to be estimated from the data.

### 4.2.2 Data preparation

427 We prepare the vector `y` containing the observations for the response variable as well as the design
428 matrix `X`:

```
y = log(air.data[,1])
X = as.matrix(air.data[,-c(1,5,6)])
X = cbind(poly(air.data[,2], degree=2),poly(air.data[,3], degree=2),poly(air.data[,4], degree=2))
```

### 4.2.3 OLS estimation

430 We first estimate the model with the ordinary least squares (OLS) approach, using bot the full dataset
431 and the one obtained by removing the outlier:

```
ols.full = lm(y~X)
ii = which(y<1)
ols = lm(y[-ii]~X[-ii,])
results = cbind(ols.full$coefficients,ols$coefficients)
print(results)
```

```
                        [,1]       [,2]
(Intercept)  3.4159273  3.4361465
X1           2.6074060  2.2885974
X2          -1.2034322 -0.8487403
X1          -2.2633108 -2.5209875
X2           1.2894833  1.1009511
X1           4.2120018  3.9218225
X2           0.4501812  0.8936152
```

As expected, the OLS estimates are sensitive to the presence of outliers in the data. In particular, we observe that the unique outlier in the data has a non-negligible impact on the estimated regression coefficient of the variables $(\texttt{Temp})^2$ and $(\texttt{Solar.R})^2$.

### 4.2.4 MMD estimation with the default settings

Using the default settings of the `mmd_reg` function, the computationally cheap estimator $\tilde{\theta}_{\text{reg}}$ with a Gaussian kernel $k_Y(y, y')$ is used, and the model is estimated as follows:

```
mmd.tilde = mmd_reg(y,X)
summary(mmd.tilde)
```

```
====================== Summary ======================
Model:              linearGaussian
Estimator:          theta tilde (bdwth.x=0)
-----------------------------------------------------
  Coefficients          Estimate
-----------------------------------------------------
  (Intercept)              3.427
  X1                      2.3448
  X2                     -0.8132
  X3                      -2.329
  X4                      1.0565
  X5                      4.1788
  X6                      0.8369
-----------------------------------------------------
  Std. dev. of Gaussian noise : 0.4484 (estimated)
-----------------------------------------------------
  Kernel for y: Gaussian with bandwidth 0.5821
=====================================================
```

As expected from the robustness properties of MMD based estimators, we observe that the estimated values of the regression coefficients are similar to those obtained by OLS on the dataset without the outlier. The `summary` command also returns the value of the bandwidth parameter $\gamma_Y$ obtained with the median heuristic and used to compute the estimator.

To estimate the model using the alternative estimator $\hat{\theta}_{\text{reg}}$ we need to choose a non-zero value for `bdwth.x`. Using the default setting, this estimator is computed as follows:

```
mmd.hat = mmd_reg(y,X,bdwth.x="auto")
summary(mmd.hat)
```

```
====================== Summary ======================
Model:              linearGaussian
Estimator:          theta hat  (bdwth.x>0)
```

```
     -----------------------------------------------------------
      Coefficients             Estimate
     -----------------------------------------------------------
      (Intercept)               3.4269
      X1                        2.3444
      X2                       -0.8131
      X3                       -2.3297
      X4                        1.056
      X5                        4.1786
      X6                        0.8374
     -----------------------------------------------------------
      Std. dev. of Gaussian noise : 0.4482 (estimated)
     -----------------------------------------------------------
      Kernel for y: Gaussian with bandwidth 0.5821
      Kernel for x: Laplace with bandwidth 0.0215
     ===========================================================
```

We remark that the value for `bdwth.x` selected by the default setting is very small, and thus the two MMD estimators provide very similar estimates of the model parameters.

### 4.2.5   Tuning the fit in MMD estimation

Above, the estimator $\hat{\theta}_{\text{reg}}$ was computed using a Laplace kernel for $k_X(x, x')$ and a Gaussian kernel $k_Y(y, y')$, and using a data driven approach to select their bandwidth parameters $\gamma_X$ and $\gamma_Y$. If, for instance, we want to use a Cauchy kernel for $k_X(x, x')$ with bandwidth parameter $\gamma_X = 0.2$ and a Laplace kernel for $k_Y(y, y')$ with bandwidth parameter $\gamma_Y = 0.5$, we can proceed as follows:

```
mmd.hat = mmd_reg(y,X,bdwth.x=0.1,bdwth.y=0.5,kernel.x="Cauchy",kernel.y="Laplace")
summary(mmd.hat)
```

```
====================== Summary ======================
Model:                linearGaussian
Estimator:            theta hat  (bdwth.x>0)
-----------------------------------------------------------
  Coefficients             Estimate
-----------------------------------------------------------
  (Intercept)               3.4106
  X1                        2.4541
  X2                       -0.8421
  X3                       -2.3274
  X4                        1.1086
  X5                        4.2962
  X6                        0.9091
-----------------------------------------------------------
  Std. dev. of Gaussian noise : 0.4483 (estimated)
-----------------------------------------------------------
  Kernel for y: Laplace with bandwidth 0.5
  Kernel for x: Cauchy with bandwidth 0.1
===========================================================
```

We recall that different choices for the kernels $k_Y(y, y')$ and $k_X(x, x')$ lead to different MMD estimators, which explains why the estimates obtained here are different from those obtained in Section 4.2.4.

### 4.3    Robust Poisson regression

#### 4.3.1    Dataset and model

As a last example we consider the same dataset and task as in Section 4.2, but now assume the
following Poisson regression model:

$$\texttt{Ozone} \sim \text{Poisson}\Big( \exp \big( \alpha + \beta_1 \texttt{Temp} + \beta_2 (\texttt{Temp})^2 + \beta_3 \texttt{Wind} + \beta_4 (\texttt{Wind})^2$$

$$+ \beta_5 \texttt{Solar.R} + \beta_6 (\texttt{Solar.R})^2 \big) \Big).$$

Noting that the response variable is now `Ozone` and not its logarithm as in Section ??, we modify the
vector y accordingly:

```
y = air.data[,1]
```

In the histogram above we observe that there is one isolated observation (i.e.~outlier), for which the
observed value of `Ozone` is larger than 150.

#### 4.3.2    GLM estimation

We start by estimating the model with the generalized least squares (GLS) approach, using both on
the full dataset and the one obtained by removing the outlier:

```
glm.full = glm(y~X,family=poisson)
ii = which(y>150)
glm = glm(y[-ii]~X[-ii,],family=poisson)
results = cbind(glm.full$coefficients,glm$coefficients)
print(results)
```

```
                  [,1]          [,2]
(Intercept)  3.5168945   3.506950377
X1           2.4621470   2.332854309
X2          -0.8796456  -0.827578293
X1          -2.4753279  -2.167010600
X2           0.9498002   0.559589428
X1           4.0664346   4.289933463
X2          -0.3247376  -0.003300254
```

It is well-known that the GLM estimator is sensitive to outliers and, as a result, we observe that the
unique outlier present in the data has a non-negligible impact on the estimated regression coefficients
of the model.

#### 4.3.3    MMD estimation with default setting

We first estimate the model parameter using the estimator $\tilde{\theta}_{\text{reg}}$. Using the default setting of `mmd_reg`
this is done as follow:

```
mmd.tilde = mmd_reg(y,X,model="poisson")
```

Warning in mmd_reg(y, X, model = "poisson"): Warning: The maximum number of
iterations has been reached.

```
summary(mmd.tilde)
```

```
======================= Summary =======================
Model:                poisson
Estimator:            theta tilde (bdwth.x=0)
-------------------------------------------------------
   Coefficients          Estimate
-------------------------------------------------------
   (Intercept)             3.3754
   X1                      2.4766
   X2                     -1.0967
   X3                     -2.3401
   X4                      0.3289
   X5                      4.7545
   X6                      0.0784
-------------------------------------------------------
-------------------------------------------------------
   Kernel for y: Laplace with bandwidth 18.3848
=======================================================
```

As for the linear regression example, we observe that the estimated values of the regression coefficients are similar to those obtained by GLM on the dataset without the outlier data point.

Finally, we estimate the model parameters using the estimator $\hat{\theta}_{\text{reg}}$ with its default setting:

```
mmd.hat = mmd_reg(y,X,model="poisson",bdwth.x="auto")
```

```
Warning in mmd_reg(y, X, model = "poisson", bdwth.x = "auto"): Warning: The
maximum number of iterations has been reached.
```

In this case, we obtain a warning message indicating that the maximum number of iterations `maxit` allowed for the optimization algorithm has been reach. The value of `maxit`, set by default to $50,000$, can be increased using the `control` argument of `mmd_reg`. For instance, setting `maxit=10^6` remove the warning message:

```
mmd.hat = mmd_reg(y,X,model="poisson",bdwth.x="auto",control=list(maxit=10^6))
summary(mmd.hat)
```

```
======================= Summary =======================
Model:                poisson
Estimator:            theta hat  (bdwth.x>0)
-------------------------------------------------------
   Coefficients          Estimate
-------------------------------------------------------
   (Intercept)              3.368
   X1                      2.4909
   X2                     -1.0501
   X3                      -2.386
   X4                      0.3063
   X5                      4.7321
   X6                     -0.0961
-------------------------------------------------------
-------------------------------------------------------
   Kernel for y: Laplace with bandwidth 18.3848
   Kernel for x: Laplace with bandwidth 0.0215
```

As for the linear regression model, the value of `bdwth.x` selected by the data driven approach implemented in `mmd_reg` is close to zero and the estimate values of the model parameters are therefore similar to those above obtained with the estimator $\tilde{\theta}_{\text{reg}}$.

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

# Session information

```
R version 4.4.1 (2024-06-14)
Platform: x86_64-pc-linux-gnu
Running under: Ubuntu 24.04.2 LTS

Matrix products: default
BLAS:   /usr/lib/x86_64-linux-gnu/blas/libblas.so.3.12.0
LAPACK: /usr/lib/x86_64-linux-gnu/lapack/liblapack.so.3.12.0

locale:
 [1] LC_CTYPE=C.UTF-8       LC_NUMERIC=C           LC_TIME=C.UTF-8
 [4] LC_COLLATE=C.UTF-8     LC_MONETARY=C.UTF-8    LC_MESSAGES=C.UTF-8
 [7] LC_PAPER=C.UTF-8       LC_NAME=C              LC_ADDRESS=C
[10] LC_TELEPHONE=C         LC_MEASUREMENT=C.UTF-8 LC_IDENTIFICATION=C

time zone: Etc/UTC
tzcode source: system (glibc)

attached base packages:
[1] stats     graphics  grDevices datasets  utils     methods   base

other attached packages:
[1] regMMD_0.0.1

```

```
loaded via a namespace (and not attached):
 [1] compiler_4.4.1    fastmap_1.2.0     cli_3.6.2         htmltools_0.5.8.1
 [5] tools_4.4.1       rmarkdown_2.29    knitr_1.50        jsonlite_1.8.9
 [9] xfun_0.52         digest_0.6.37     rbibutils_2.3     Rdpack_2.6.1
[13] rlang_1.1.4       renv_1.1.4        evaluate_1.0.0
```

