# OpenReview forum: "regMMD: an R package for parametric estimation and regression with maximum mean discrepancy"
_Computo — Accepted by Computo_

### Review · Reviewer_Z78c · 2025-08-01

**Summary Of Contributions:**

This paper introduces an R package developed for performing minimum Maximum Mean Discrepancy (MMD) estimation in various parametric inference and regression settings. Minimum MMD estimators are particularly robust to outliers, a property supported by theoretical guarantees in the literature. The authors provide a detailed overview of the package, including illustrative examples and accompanying code. They also present results from several analyses conducted using the package. Additionally, the package supports multiple optimisation methods, including gradient descent, stochastic gradient descent, and exact optimisation when feasible. Users can choose from a predefined set of available models.

**Audience:**

Yes

**Broader Impact Concerns:**

No Broader Impact Concerns

**Claims And Evidence:**

Yes

**Requested Changes:**

- [Critical] I noticed that when the data is continuous and a discrete model is picked (e.g. Binomial, discrete.uniform) (or vice versa), inference is performed normally, and there is no warning or error. I assume this is because there is no input validation to check if there is consistency between the model and the data type. This should be enforced as the MMD itself is not well-defined for probability measures on different data spaces.

- [Critical] Similarly, the package allows negative observations even when the chosen model is defined only for non-negative values (for example the Binomial). The authors should add a warning for such cases.

- [Critical] With respect to my comment about optimisation in the previous question, could the authors please clarify if there are any warnings for NaNs or exploding gradients? Also, I think allowing users to access logs or traces of optimisation steps would be very helpful so that they can understand why the optimisation might have failed or to compare different optimisation settings for parameter tuning purposes. I saw there is the option trace=TRUE for mmd_reg but as far as I understood by trying it there are not many details of the optimisation returned. An example of this in the paper could also be helpful.

- [Critical] In Section 3.4, could the authors include a discussion about any theoretical convergence guarantees of the optimisation algorithms for different models (or their absence)? Even a brief discussion referencing existing theoretical results (e.g., those in Section 5.3 of Cherief-Abdellatif et al., 2022) would be helpful to clarify in which cases the user can expect convergence to global minima for each of the suggested algorithms.

- [Critical] The authors should include a discussion of the limitations of the package in its current form (e.g. model choice availability, potential optimisation pitfalls) and possible future extensions, as it's normally done in other Computo software papers (e.g. https://computo-journal.org/published-202505-ferte-reservoirnet/). This would clarify the package’s position within the existing literature and help the research community extend the software to more general applications.


- [Non-critical] Given that more complicated likelihood-free models (such as simulators) are a big use case of MMD estimators, is there scope for allowing for a user-specified generator function as an input to the algorithm, such that SGD can be performed by sampling from the generator and approximating the MMD as described in Briol et al. 2019? I don’t think that’s absolutely critical, as the package/paper already offers a very useful tool for commonly used models, but I believe it would greatly strengthen the work and its impact. If not, it would be interesting to understand what are the technical difficulties associated with implementing this and why it’s not possible.

- [Non-critical] Another aspect that is not discussed is what happens when there is not a single minimum of the objective function. In that case, the minimiser of the MMD is a set rather than a single value. Could the authors comment on the behaviour of the optimisation in this case or any warnings that accompany the package in such cases?

- Typos: I have included some typos I spotted below:
   1. Line 78 (remove “to”)
   2. Line 105 (remove “%” )
   3. Lines 113 & 114 (I believe $\hat{\theta}$ should be $\hat{\theta}_{reg})
   4. Line 249 (I think “unknown” should be “known”)
   5. Line 326 (fix brackets)
   6. Line 333 (fix quotations)
   7. Line 430 (“bot”)
   8. Line 492 (spacing)
   9. Line 568 (“remove” should be “removes”)

**Strengths And Weaknesses:**

Strengths:
- The paper is clearly written and easy to follow.
- It effectively outlines the scenarios where the package is useful and the types of inference problems suitable for minimum MMD estimation.
- The authors have provided detailed instructions for reproducing the experiments and conducting new analyses using the package. I have played around with the code and tried a few examples and everything seems very straightforward to use.
- The authors address some potential warning messages from the algorithms and offer guidance on how to interpret and respond to them.
- Alongside the package documentation, the paper serves as a comprehensive user manual and is ready for immediate use.

Weaknesses:
- There is a small number of cases where warnings should be raised due to a mismatch in input data and model type which I outline below.
- From my experience, a challenging aspect of minimum MMD estimation optimisation is the problem of vanishing or exploding gradients when the kernel bandwidth is not chosen appropriately. This aspect is not discussed beyond warnings relating to maximum iterations achieved.
- There is no adequate discussion on the existence or absence of any convergence guarantees of the optimisation algorithms for MMD estimation and regression, and the impact of hyperparameter choices.
- Although the package offers a variety of commonly used models, it doesn’t offer the option of a user-defined model. This is a bit limiting as minimum MMD estimation has been particularly useful in the context of likelihood-free models, for which the analytical form of the likelihood is unavailable, as studied in Briol et al. 2019 and Cherief-Abdellatif et al. 2022 in the frequentist setting and in likelihood-free Bayesian inference, for example, in Approximate Bayesian Computation and Simulation-Based Inference (Legmanti et al. 2025, Dellaporta et al. 2022) and Generalised Bayesian inference (Pacchiardi et al. 2024) applications.

---

> ### Author Response · Authors · 2025-09-17
> **Thank you**
>
> First, we want to thank you for your detailed review and your encouraging comments! We will now address one-by-one all the critical and non-critical points you raised, and make some proposals to amend the paper and/or the package accordingly. Please kindly let us know if you find these satisfying, or if you believe another approach would be preferable.
>
> Critical point 1) [discrete vs. continuous data]
>
> This is a very critical point indeed, so we think this deserves a full discussion before we implement any changes. Of course, it is true that we should absolutely not use Gaussian models for categorical variables such as "male/female" encoded by numbers "0/1". That being said, the distinction "discrete/continuous" is not always that clear. First, very often, Gaussian noise is used in econometrics when we model wages or GDPs, that are discrete by nature (0.01$ cannot be divided).
>
> More importantly, the MMD distance $D(P,Q)$ is well defined even when $P$ is discrete (say, the Poisson distribution) and $Q$ is continuous (eg, Gaussian). Indeed, it's true that the KL divergence $KL(Q\|P) $ is infinite in this case, and thus, a likelihood-based approach makes no sense to estimate a Poisson model when the observations are not integers. But the MMD distance is well defined here, and this is not just an accident. Indeed, we know that, for large $\lambda$, the Poisson distribution $\mathcal{P}(\lambda)$ is well approximated by a Gaussian distribution $\mathcal{N}(\mu=\lambda,\sigma^2=\lambda)$. Thus, even if we have discrete but very large observations, why not fit a Gaussian model rather than a Poisson? This is in essence what is done in practice when we fit Gaussian distributions to GDPs and wages. The theoretical guarantees reminded in Theorem 2.1 do not require the model to be well specified, and actually, they allow for "extreme" misspecification where we would estimate a Poisson by a Gaussian...
>
> Another example can help to understand this: let $Q$ be a Dirac mass at $0$, and $P=\mathcal{N}(0,V)$ be a Gaussian distribution with variance $V$. As long as $V>0$, we have $KL(Q\|P)=\infty$, while $MMD(Q,P)$ is finite and will actually converge to $0$ when $V\rightarrow 0$. We can approximate discrete distributions buy continuous ones, and vice-versa. This is not a logical flaw, this is actually one of the nice properties of the MMD distance. In some situations, this is a desirable thing to do.
>
> We totally understand your concerns, but also do not want to prevent users to fit Gaussian on integer values observations in settings where this would make sense. So, we would prefer to add a discussion on this topic in the paper, and in the package description, rather than to prevent this completely.
>
> Critical point 2)
>
> We believe this comment is related to the previous one. MMD minimization is meant to provide robust estimates, and in particular, outliers do not lead the criterion to be infinite. If you fit a binomial distribution to a dataset that contains one negative entry, is it a logical flaw, or did the procedure just discard an outlier? In any case, MMD minimization does not pretend to find the truth, only to find the best binomial approximation of the true distribution of the data...
>
> Critical point 3)
>
> In mmd_reg the optimization stops when there is an NaN gradient, and an error is returned by the function. There is no warning for exploding gradients: as long as the gradient is not NaN the optimization process continue until the maximum number of steps (or convergence criterion) is reached. For mmd_reg, the trace is indeed returned, which allows the user to visually assess (component by component) how the parameter value evolves in the course of the optimization process. We agree that we should illustrate this in the paper, and will to do it in its revised version. Diagnostic convergence for noisy optimization is not an easy problem (because for most models neither the value of the objective function nor its gradient can be computed exactly), and therefore providing users with other useful information regarding the optimization process is not trivial. However, for some models we can evaluate the gradient of the objective function, and for these models we can update the package so that the  final value of the gradient is provided to the users. For the function mmd_est, a lower bound on the problematic parameters prevents the gradient to be NaN. The trace is currently not provided for this function but will be added in the revision of the package.

---

> > ### Author Response · Authors · 2025-09-17
> > **(2/2)**
> >
> > Critical point 4)
> >
> > The MMD criterion is non-convex in most models, making the theoretical analysis of gradient descent difficult. Especially, in most cases, the gradient will vanish when we are infinitely far from parameters of interest. Worse, in some models, we can expect multiple local minima. In (Cherief-Abdellatif et al., 2022), a theoretical analysis of the stochastic gradient algorithm is indeed provided, but this is in the only model we are aware of that makes the MMD criterion convex: mixture of fixed distributions, where we only estimate the proportions.
> >
> > In other models, we indeed can't provide converge guarantees to the global minimum. We tried to avoid "bad local minima" by initializing the algorithms to preliminary estimators (e.g. the median for location parameters), and obtained indeed very good results in practice. In case the convergence seems to be poor (see point 3 above), our only recommendation would be to run the algorithm multiple times with random initialization.
> >
> > You are totally right that we should include this whole discussion in the paper: we will include it in Section 3.4.
> >
> > Critical point 5)
> >
> > We agree that such a discussion should be provided in Section 3.4, and will add it in the revised version of the manuscript. It will certainly include a discussion on the convergence of the SGD algorithm, missing models, and the possibility to include confidence intervals in the future.
> >
> > Non-critical point 1)
> >
> > We agree that it would be a good idea to add likelihood free models in the package. We however keep this as a future extension/limitation of the current package (that will be made clear in the revised version of the paper, following the previous reviewer's remark), for the following reason: while MMD is used in machine learning and statistics to address challenging tasks, the main goal of our paper Alquier and Gerber 2024 is to show that MMD is useful to solve more standard statistical problems, and the goal of this R package is to encourage users to do so.
> >
> > For example, consider the generative adversarial networks (GAN) trained by (Dziugaite, Roy and Ghahramani, 2015) by MMD minimization. It's true that this model is not included in our package... but this model has no likelihood, and the optimization technique used is completely different (automatic differentiation) and would require completely different tools. While we agree it would be nice to make more software available for such models, this would definitely be a different package.
> >
> > Non-critical point 2)
> >
> > We expect that in most cases the objective will have multiple local minima. If the model is corrupted with outliers this is somehow inevitable. In this case, as for standard (S)GD algorithms, we expect the optimization to converge to a local minima located near the initial value. Hence, the choice of the latter is critical, and in the package  the optimizer (generally) uses, by default, the maximum likelihood estimator (MLE) as starting value. The rational for this choice is the following: even if the observations are corrupted by outliers, we expect the minimizer of the MMD to be, in most cases, not too far from the MLE. Of course, this is an imperfect strategy, but a reasonable default one (in our opinion). We however agree that this should be discussed in the revised version of the paper.
> >
> > Typos:
> >
> > Thank you for spotting these typos, we will fix all of them.

---

> ### Comment · Reviewer_Z78c · 2025-10-02
>
> Thank you very much for the detailed response to my review. Regarding points 1-2, thank you very much for clarifying. I totally agree that it makes sense not to prevent users from using models with different "mismatched" data types in settings where this makes sense, and I hadn't thought of some of these cases, so I agree that there is no need for any changes regarding the discrete/continuous case.
>
> My original thought (perhaps not totally clear) was to include a warning rather than preventing the user from using the method altogether. At least for point 2 (say we have negative values for a Bernoulli example), although the MMD is a robust distance and would not blow up in the presence of an outlier, the user would be indirectly defining a kernel $k$ on $\mathcal{X} \times$ {0,1} and then using an empirical measure which is *not* over this same space, as an input to the MMD. I agree that the point of the MMD is not to detect outliers or model misspecification but to simply find the best approximation to the true distribution inside the model family. However, I am wondering whether a simple warning would be helpful in cases where this could be due to a bug? I would be interested in hearing your thoughts.
>
> Regarding the rest of the points, I am happy with all the responses, and I am glad to hear all the relevant discussions will be added to the paper.

---

### Review · Reviewer_6Nnd · 2025-08-17

**Summary Of Contributions:**

The maximum mean discrepancy (MMD) is a class of kernel-based loss functions that are equipped with appealing computational and theoretical properties. Estimating a model parameter based on minimisation of the MMD has increasingly become a popular approach to statistical inference because of the strong robustness again outlier and adversarial contamination. However, to date, there has been no standard software implementation to compute the minimum MMD estimator for one's applications. This paper introduces the R package, regMMD, that computes the minimum MMD estimator of a variety of models for density estimation and regression problems. Along with the implementation, the paper provides preliminaries on computational and theoretical properties of minimum MMD estimators.

**Audience:**

Yes

**Claims And Evidence:**

Yes

**Requested Changes:**

I would personally be happy to recommend the acceptance with no major change to the paper.

**Strengths And Weaknesses:**

The paper is very well written and easy to follow, recapping important preliminaries on computational and theoretical properties of the MMD at the right level of granularity for a broad audience. The implemented software is easy to use, with a set of example use cases detailed throughly within the paper. There seems no major weakness that particularly needs to be addressed for publication. I suppose that models in the package have to be predefined ones, and it would be nice to clearly mention whether it is possible for users to plug their own models in the main functions, as some readers may wonder. If there is any plan for further development of the package in the future, adding functions for statistical tests based on the MMD and computation of the asymptotic covariance may also be interesting as they would be useful and lightweight additions, but these would not be necessary for the current version.

---

> ### Author Response · Authors · 2025-09-17
> **Thank you**
>
> We thank you for your very positive comments and encouraging feedback on our work.
>
> We agree that user-defined likelihood and confidence intervals would be necessary for some applications. We definitely keep them in mind, and hope to be able to implement them in the future.

---

### Comment · Action_Editor_K68L · 2025-09-08
**Rebuttal period**

Dear authors,

We have received two reports for your submission to Computo entitled "regMMD: an R package for parametric estimation and regression with maximum mean discrepancy".

A rebuttal period of 6 weeks has started on August 17th: this is intended to allow for discussion with the referees before they issue a final opinion. During this period, you can make any changes to your submission that you feel are necessary and that you are able to make. At the end of this period, a decision will be made, ranging from final acceptance to more substantial requests for modification.

Best regards

---

> ### Author Response · Authors · 2025-09-17
> **Thank you**
>
> Dear Action Editor,
>
> Thank you! We uploaded a reply to each reviewer and discussed all the points they raised. We proposed solutions to the reviewers and will implement them immediately if they find these satisfactory.

---

### Comment · Action_Editor_K68L · 2025-10-03
**End of rebuttal period**

Dear authors and reviewers,

I would like to thank you all for the thorough work conducted on this paper. The points raised by the reviewers have been carefully discussed during the rebuttal period, and the authors have provided satisfactory responses, either by justifying certain aspects or by correcting others. Both reviewers agree that the proposed strategy to amend the paper is appropriate. On this basis, I am confident that the manuscript will be suitable for acceptance once the requested revisions have been completed.
I encourage the authors to update the submission on the GitHub repository at their earliest convenience.

---

> ### Author Response · Authors · 2025-10-22
> **Revision ready**
>
> Dear Action Editor,
>
> We updated the package to include:
> - the warning for continuous data in discrete models,
> - the trajectories of gradient descents.
> The new version of the package (0.0.3) is now on CRAN: https://cran.r-project.org/web/packages/regMMD/index.html
>
> We also updated the paper accordingly: https://pierrealquier.github.io/regMMD-paper/
> and we inserted the discussions required by the reviewers. More specifically:
>
> # Critical points 1)-2)
> For the function mmd_est, a warning has been added in case continuous observations are used to fit a discrete model, and explanations why doing so is allowed in the context of MMD estimations have been added in Section 3.1 of the paper. The function mmd_reg requires the observations for the response variable to be in the support of the model.
>
> #  Critical point 3)
> The function mmd_est has been modified so that the users can access the trajectory of the sequence of iterations. An example showing how to do so has been added (in Section 4.1)
>
> # Critical points 4)-5)
> Explanations why the optimisation of the MMD criterion is challenging have been added in Section 3.4
>
> # Critical point 6)
> A conclusion has been added (Section 5), we here we notably highlighting  what the main weaknesses of the current version of the package.
>
> We hope you will find this version suitable for publication.

---

> > ### Author Response · Authors · 2025-11-07
> >
> > Dear AE,
> > Sorry about that, but I'm not really sure what the next step is... We updated the package according to the requests of the reviewers (see our previous message) and the paper.

---

### Comment · Action_Editor_K68L · 2025-10-10
**Official recommendation**

Dear reviewer,

I would like to thank you again for your thorough work on this paper. In order to move forward with the review process, we would need you to provide an official recommendation (which differs from the official comment you made). I understand that you still have some concerns regarding one specific aspect, please feel free to recommend whatever suits you.

Could you please indicate your official recommendation through the system ?

Thanks a lot

Marie-Pierre Etienne, as associate editor.

---

### Comment · Action_Editor_K68L · 2025-11-07
**The production phase of your article is about to start**

Dear author,

Congratulations on the acceptance! Before the production phase of your paper starts, we need you to perform a couple of steps.

* Can you please check that your affiliation and metadata are correctly filled in in the header of your article (name, affiliation, and url of each author).
* Please run the CI and fix any remaining issues with the build.
* Be sure to use the latest Computo extension
* Finally, transfer the ownership of the github repository to computo. See the Authors guidelines for a small description on how to perform the ownership transfer. In order to do so, you have been temporarily added to the computo organization.

---

### Comment · Action_Editor_K68L · 2025-11-07
**The production phase of your article is about to start**

Dear authors,

Congratulations on the acceptance! Before the production phase of your paper starts, we need you to perform a couple of steps.

 * Be sure to use the latest Computo extension,
 * Check that your affiliation and metadata are correctly filled in in the header of your article (name, affiliation, and url of each author).
 * Please run the CI and fix any remaining issues with the build.
 * Finally, transfer the ownership of the github repository to computo. See the Authors guidelines for a small description on how to perform the ownership transfer. In order to do so, you have been temporarily added to the computo organization.

---

### Note · Reviewer_6Nnd · 2025-10-01

**Comment:**

I would recommend the acceptance as in the previous review. I found the paper well-written with technical contents explained at the right level of granularity for a broad audience. The examples presented in the paper demonstrated to me that the implemented software is easy to use. Beyond the example case studies, the other review pointed out a few technical points regarding the implementation, on which I believe the authors will clarify or improve in the paper.

**Audience:**

Yes

**Claims And Evidence:**

Yes

**Decision Recommendation:**

Accept

---

### Note · Reviewer_Z78c · 2025-10-10

**Comment:**

As indicated in the rebuttal, I think the paper is a great fit for the journal and recommend acceptance of the paper.

**Audience:**

Yes

**Claims And Evidence:**

Yes

**Decision Recommendation:**

Accept

---

### Decision · Action_Editor_K68L · 2025-11-07

**Recommendation:** Accept as is

**Comment:**

Dear authors,

After reading the  updated version of your paper which addresses the requests raised by the reviewers, i am plase to inform you that your paper is accepted.

**Audience:**

This paper is relevant for Computo's readership.

**Claims And Evidence:**

Claims And Evidence:

Yes

---

> ### Decision · Editors_In_Chief · 2025-11-07
>
> I approve the AE's decision.